# Benchmarking Quantum Computer Simulation Software Packages: State Vector Simulators

**Amit Jamadagni**[1,2*]**, Andreas M. Läuchli**[1,3]**, and Cornelius Hempel**[2,4]

[1]Laboratory for Theoretical and Computational Physics, Paul Scherrer Institute (PSI), 5232 Villigen, Switzerland
[2]ETH Zürich-PSI Quantum Computing Hub, Paul Scherrer Institut, 5232 Villigen, Switzerland
[3]Institute of Physics, Ecole Polytechnique Fedérale de Lausanne (EPFL), 1015 Lausanne, Switzerland
[4]Institute for Quantum Electronics, ETH Zürich (ETHZ), 8093 Zürich, Switzerland
[*]Present address: Leibniz Supercomputing Centre, 85748 Garching, Germany

## ABSTRACT

Rapid advances in quantum computing technology lead to an increasing need for software simulators that enable both algorithm design and the validation of results obtained from quantum hardware. This includes calculations that aim at probing regimes of quantum advantage, where a quantum computer outperforms a classical computer in the same task. High performance computing (HPC) platforms play a crucial role as today's quantum devices already reach beyond the limits of what powerful workstations can model, but a systematic evaluation of the individual performance of the many offered simulation packages is lacking so far. In this Technical Review, we benchmark several software packages capable of simulating quantum dynamics with a special focus on HPC capabilities. We develop a containerized toolchain for benchmarking a large set of simulation packages on a local HPC cluster using different parallelisation capabilities, and compare the performance and system size-scaling for three paradigmatic quantum computing tasks. Our results can help finding the right package for a given simulation task and lay the foundation for a systematic community effort to benchmark and validate upcoming versions of existing and also newly developed simulation packages.

## Key Points

- Efficient, general purpose software simulators for quantum computation provide the means to develop algorithms, anticipate performance gains and estimate resource requirements for future deployment on actual quantum hardware.

- A large variety of software packages designed to simulate quantum computers exist, but not all are regularly maintained or continue to be developed.

- Depending on the simulation task, the computational performance of packages can differ by more than two orders of magnitude at both small and large problem sizes, even when run on the exact same hardware.

- Hardware accelerations enabled by multithreading, the use of GPUs yield substantial performance improvements — yet at problem sizes between 25 qubits (CPUs) and 30 qubits (GPUs), all evaluated packages cross over into exponential scaling behavior, however we observe significant differences in pre-factors.

## Introduction

The rapidly growing interest in quantum computers from both the academic and commercial side fuels the need for software packages that can simulate their operation and aid both hardware and algorithm development. These software environments generally model the quantum state evolution of an isolated set of qubits acted on by a set of well-defined quantum gate operations with an intrinsic error rate set only by the classical computer's numerical precision. Some packages further support an open quantum systems simulation, which can also model external perturbations, thereby allowing for the simulation of actual, noisy quantum hardware at the expense of additional computational overhead. In all cases, the available computational hardware, the underlying mathematical formalism[1] and its particular software implementation place stringent limits on the system sizes that can be explored. With the exception of a few special cases, even High Performance Computing (HPC) platforms often cannot reach beyond a system size of about 50 simulated qubits before they run into resource limitations[2–5]. Yet, even at moderate scales of a few tens of qubits, efficient simulation with reasonable execution times is a formidable challenge due to the vast amounts of data that have to be processed to faithfully store and manipulate a generic quantum state. With an amount scaling exponentially as $2^{\#\text{qubits}}$, memory and the transfer of data between processing and storage are at the core of the slow-down in computation speed and this is where HPC architecture and a tailored software implementation can make a difference.

In this Technical Review, we benchmark a number of locally-installable quantum computing simulation packages on a HPC cluster in a standardized way using three exemplary quantum computational tasks, which are among some of the most important primitives in quantum algorithms[6]. We start with a brief survey of available software packages grouped by simulation strategies, and then down-select to a list of 24 packages. The chosen packages are integrated into a containerized toolchain workflow, which ensures performance evaluation on equal footing as well as extensibility, reproducibility and ease

of maintenance. For the three circuit-based computational tasks – a gate-based simulation of Heisenberg spin dynamics, random circuit sampling and a quantum Fourier transform circuit – we investigate the wall-clock time as a function of the number of simulated qubits, desired numerical precision and the specific HPC capabilities enabled. In all settings, we perform a ranking in terms of the fastest package in the exponential scaling limit and also compare the scaling behavior and overheads at intermediate problem sizes.

## Quantum computer simulation packages

The recent advances in quantum computing hardware has led to increased momentum in the development of software that can simulate quantum circuits, with many new software packages appearing over the past 5 years. The development goals of these packages range from the validation of quantum hardware using error models on an abstract or a device-specific level, to the simulation of hardware-agnostic high-level quantum logic operations that aim to efficiently reach as far as possible to the boundary of quantum advantage, where classical computers can no longer simulate a quantum system's dynamics in acceptable runtimes. HPC clusters are naturally positioned at the forefront of this boundary and therefore a key aspect of our analysis is to benchmark the performance of HPC-compatible software packages.

Several different strategies are employed by software packages in the simulation of quantum circuits, usually chosen based on the primary use case envisioned by the developers. Table 1 provides a non-exhaustive list of packages grouped by their underlying simulation approach with links to a corresponding publication or website. Further lists of packages can be found on the websites Quantiki and the Quantum Open Source Foundation.

First, there are statevector-based simulators that simulate the evolution of pure quantum states. Here, a statevector of $N$ qubits is represented by a $2^N$-sized vector of complex valued entries which are modified under the action of a formally $2^N \times 2^N$-sized unitary matrix operator. A more complex (and resource intensive) simulation is offered by the density matrix formalism, which further allows to simulate mixed states, which usually arise under the influence of noise and in the case of open quantum systems. Tensor networks offer an alternate representation of the quantum state. In the example of Matrix Product States (MPS) or Projected Entangled Pair States (PEPS)[68, 69] the wavefunctions are expressed in a compressed way using a set of local tensors to be contracted. The unitaries are represented as Matrix Product Operators (MPO) or Projected Entangled Pair Operators (PEPO), which are smaller in size than the above unitaries. However, with increasing entanglement in the system, the size of the tensors, referred to as the so-called bond dimension, is forced to grow to maintain an accurate approximation. Clifford algebra based simulators have been developed[70] that allow the access of qubit numbers on the order of a million and more. They are mainly used to investigate quantum error correction codes[52]

**Table 1.** List of quantum computing simulation packages grouped by supported simulation approach.

| **Statevector simulators** |
| --- |
| qiskit[7], cirq[8], qsimcirq[9], pennylane[10], pennylane lightning[10], qibo[11], qibojit[12], Intel QS[13], projectq[14], qrack[15], qpanda[16], qcgpu[17], quest[18], qulacs[19], qpp[20], SV-Sim[21], Yao[22], HiQ[23], HybridQ[24], Braket, myQLM, QuTiP[25], PyQuil[26], pytket[27], Microsoft QDK Simulator, Blueqat (qaqarot), Quantastica Toaster, HyQuas[28], MPIQulacs[29], JUQCS[30], Quimb[31], NVidia cuQuantum[32], Spinoza[33], QuantumFlow, QPlayer[34], Torchquantum[34], pyqtorch, QCompute, QX Simulator, Basiq[35], MIMIQ (QPerfect), Qristal, QCLAB++[36], quantum-gates[37], Qrisp |
| **Density matrix simulators** |
| qiskit, cirq, pennylane, qibo(jit), braket, hybridq, intelqs, myqlm(py), qpanda, qsimcirq, quest, qulacs, q++, sv-sim, yao, quantum-gates, QuantumSim[38], NWQSim[21,39], QuaC, QuTiP (PIQS), OpenQuantumTools.jl[40] |
| **Tensor Network** |
| qiskit, PastaQ.jl[41], NVidia cuQuantum, QXTools[42], Blueqat, Tai Zhang Simulator[43], qFlex[44], HybridQ, ExaTN[45] (with TNVQM Accelerator), Jet[46], Quimb[47], TensorCircuit[48], QTensor[49], Tensorly[50], TenPy[51], MIMIQ (QPerfect), qrack |
| **Clifford gate** |
| qiskit, cirq, qrack, Stim[52], QuantumClifford.jl, PyZX[53], MIMIQ (QPerfect), pennylane |
| **Platform specific packages** |
| Strawberry Fields[54] (photons), Generic Tensor Networks[55] (neutral atoms), QSimulator.jl (superconducting qubits), IonSim.jl (trapped ions), Bloqade.jl[56] (neutral atoms), Perceval[57] (photons), isQ[58] (superconducting qubits) |
| **Other simulators or domain-specific packages** |
| OpenFermion[59] (fermionic systems, incl. chemistry), XACC[60] (multi-architecture framework), MQT DDSIM (decision tree diagrams), qrack (Decision-tree), qrack (optimized tensor networks), Interlin-q (distributed/networked QC), Tequila[61] (VQE chemistry centric), Paddle Quantum(QML)[62], MindQuantum (HiQ based, VQE + QML)[63], TensorFlow Quantum[64] (graph based, built on TensorFlow), OpenQAOA[65] (Simulators for OpenQAOA), Krotov[66] (Optimal control), Quandary[67] (Optimal control using distributed computing) |

and more recently random unitary circuits. Yet, these have been shown to not be universal for quantum computation[70] and adding non-Clifford operations to these simulations, while extending their range, rapidly makes them far less efficient. Both approaches are the leading methods in exploring large scale simulations at the edge of exact verifiability[71, 72].

Further, there are simulators that are tailored to a specific hardware architecture or application domain, aiming to em-

ulate the corresponding platform exactly or being optimized for a specific use-case.

In this manuscript, we restrict our analysis to statevector simulators. Furthermore, our selection is based on additional criteria, such as, whether a package offers to exploit the computational power provided by HPC platforms, is actively maintained and developed (gauged in terms of release cycles) and provides access to relevant documentation (feature snippets, examples and possibly use-case tutorials). Table 2 presents our down-selected list of packages benchmarked in this work and their supported HPC capabilities[*], in addition to the support for additional features like shot simulation, noise modelling and variational quantum algorithms.

## Challenges in benchmarking

There are a number of challenges that arise when profiling the performance of simulation software packages for a direct comparison on the same hardware. Firstly, a key difference among the packages is their higher level instruction set, i.e., the specific commands required to enact a particular gate operation on a given quantum state. Hence, each package requires a recasting of the same algorithmic logic into a different set of commands and parameters. Necessarily done in a manual way and bound by a package's level of mathematical abstraction, this step is time consuming and not necessarily fool-proof such that some of the instructions of the original quantum algorithm might be mistranslated.

A related challenge lies in the more frequent release cycles, especially of the newer packages, which tend to update gate definitions or add new ones informed by the development of new hardware. For instance, the package Qibo added support for the fSim gate in one of its releases, which arose as the best working gate operation on one of Google's newer quantum processors. Another example is the package Qiskit which redefined its $U2$ and $U3$ gates to adapt itself to the new definitions of the OpenQASM language V3[74]. An ideal comparison would require the same gate definitions across packages and different releases, which is hardly feasible due to additional overhead.

Another difficulty arises from package dependencies on various external libraries that have to be installed in parallel. Here, specific combinations of versions are required, which differ between packages and even releases of the same package. Hence, the resulting version conflicts generally prevent parallel installations of different packages and releases.

Given that our focus is on benchmarking packages on the exact same HPC hardware in different configurations, we further need to set the appropriate options not only in a package's configuration, but also in the job file that configures the HPC cluster for each computational task. Finally, we need to store the numerical results produced by each package for output validation and extract the profiling data in a form that allows for comparisons on equal footing.

---

[*]While we list the MPI capability, we do not present MPI benchmarks in this work.

## Containerized toolchain workflow

To address the above challenges and ensure reproducibility as well as extensibility, we have created a containerized toolchain to implement our benchmarks. Containers are virtualized images of an operating system (OS) that share the kernel resources of the underlying host. They have not only led to swift deployment of applications without the need to customize local OS installations but also to a more efficient, robust and secure use of server resources. Another key feature of containers is their portability, allowing fully installed, customized packages to be run on different server hardware without repeated installation effort.

Various containerization software has been developed and adopted by various HPC centers world-wide, e.g., Singularity[75] at Lawrence Berkeley National Laboratory and Charliecloud[76] at Los Alamos National Laboratory. Contrary to what one might expect, there is almost no performance penalty associated with this type of virtualization when compared with "bare metal" installations, which others[77] and we ourselves confirmed in experiments. The modularity of the toolchain allows for easy integration of new packages as well as comparisons of same package with different options.

### *Technical description of the implementation*

We realize our toolchain workflow with a number of Ubuntu Singularity container images into which packages are installed jointly with their required auxiliary libraries. All packages which use only CPU features are bundled into one container, while those that support GPUs are installed into a separate container. In addition, for MultiGPU benchmarks we have used a container maintained by the NVidia developers[78]. The toolchain, illustrated in Fig. 1, automatically generates the necessary run files for benchmarking the various combinations shown in Tab. 2, which includes the package-appropriate high level instruction set translated from the original QASM input (Fig. 2), the job scripts with the appropriate compute capability options, and other auxiliary files linking the translated files to the specific package version installed on the container.

As the packages are written a variety of languages including C, C++, Python and Julia (cf. Tab. 2), we adopt different strategies to ensure non-overlapping environments. For the python packages, we install Miniconda on the container and create separate Conda environments into which packages are installed. For Julia, we redefine the path where the libraries are installed, while also scoping it into the environment of the container. For C/C++ implementations we build libraries directly on the container and link them accordingly while generating the respective binaries/executables of the translated run files. While most of the packages are readily deployable in this way, a few packages show exceptions under some combination of HPC modality and precision. They have been marked in Tab. 2 and were hence deployed natively on the HPC environment.

**Table 2.** List of benchmarked quantum simulation packages and supported features

| Package | Language | Version | ST | | MT | | GPU | | m-GPU | | MPI | Shot | Noise | VQA[†] |
|---|---|---|---|---|---|---|---|---|---|---|---|---|---|---|
| | | | SP | DP | SP | DP | SP | DP | SP | DP | | | | |
| Braket[k] | Python | 1.38.1 | | ✓ | | ✓ | | | | | | ✓ | ✓ | ✓ |
| Cirq[8] | Python | 1.1.0 | ✓ | ✓ | | | | | | | | ✓ | ✓ | ✓ |
| cuQuantum [j]73 | Python | 22.11.0 | | | | | ✓ | ✓ | ✓ | ✓ | | ✓[m] | ✓[m] | ✓[m] |
| HiQ[23] | Python | 0.0.1 | | ✓ | | ✓ | | | | | ✓[b] | ✓[n] | | |
| HybridQ[24] | Python | 0.8.2 | ✓ | | ✓ | | ✓ | | | | | ✓ | ✓ | |
| Intel-QS(cpp)[13] | C++ | 2.1.0 | ✓ | ✓ | ✓ | ✓ | | | | | ✓[d] | ✓[o] | ✓ | ✓ |
| myQLM (py) | Python | 1.7.3 | | ✓ | | | | | | | | ✓ | ✓ | ✓ |
| myQLM (cpp)[l] | C++ | 0.0.5 | ✓ | ✓ | ✓ | ✓ | ✓[l] | ✓[l] | | | | | | |
| Pennylane(py)[10] | Python | 0.28.0 | ✓ | ✓ | | | | | | | | ✓ | ✓ | ✓[p] |
| Pennylane(cpp)[10] | C++ | 0.28.2 | ✓ | ✓ | ✓ | ✓ | ✓[c] | ✓[b] | | | | | | |
| Projectq[14] | Python | 0.8.0 | | ✓ | | ✓ | | | | | | ✓ | | |
| Qcgpu[17] | Python | 0.1.1 | | | | | ✓ | | | | | ✓ | | |
| Qibo[11] | Python | 0.1.11 | ✓ | ✓ | ✓ | ✓ | | | | | | ✓ | ✓ | ✓ |
| Qibojit[12] | Python | 0.0.7 | ✓ | ✓ | ✓ | ✓ | ✓ | ✓ | ✓ | ✓ | | ✓ | ✓ | ✓ |
| Qiskit[7] | Python | 0.41.0 | ✓ | ✓ | ✓ | ✓ | ✓ | ✓ | ✓ | ✓ | ✓[a] | ✓ | ✓ | ✓ |
| QPanda[16] | Python | 3.7.16 | | ✓ | | ✓ | | ✓[h] | | | | ✓ | ✓ | ✓ |
| Qrack[15] | C++ | 8.2.2 | ✓ | ✓ | ✓ | ✓ | ✓ | ✓ | | | | ✓[q] | ✓[r] | |
| Qsimcirq[9] | Python | 0.15.0 | ✓ | | | ✓ | ✓[b] | | ✓ | | | ✓ | ✓ | |
| Quest[18] | C | 3.5.0 | ✓ | ✓ | ✓ | ✓ | | ✓ | | | ✓[d] | ✓ | ✓ | |
| Qulacs[19] | Python | 0.5.7 | | ✓ | | ✓ | | ✓[e] | | | ✓[f] | ✓ | ✓ | ✓ |
| Q++[20] | C++ | 4.0.1 | | ✓ | | ✓ | | | | | | ✓ | ✓ | |
| SV-Sim[21] | Python | 1.0 | | ✓ | | ✓ | | | ✓[g] | ✓[d] | | ✓ | ✓[s] | ✓[s] |
| Yao[22] | Julia | 0.8.6 | ✓ | ✓ | ✓ | ✓ | ✓[d] | ✓[d] | | | | ✓ | ✓ | ✓ |

ST - Singlethread, MT - Multithread, Shot - Shot simulation/Measurement simulation, Noise - Support for noisy simulations,
VQA - Variational Quantum Algorithms, SP/DP - Single/Double Precision, (m)-GPU - (multi) Graphical Processing Unit, MPI - Message Passing Interface,
[†] We checkmark packages that offer modules with the statevector-optimizer integration,
[a] Installation difficulty: Compilation error, [b] Native HPC and container version different: native HPC run,
[c] Runtime error, [d] Native HPC and container version same: native HPC run, [e] GPU used is different from A100: GeForce RTX 2080 Ti,
[f] Closed source: not included in the benchmarks, [g] Installation difficulty: nvshmem missing while compilation, [h] No result, after time limit reached
[j] Using *qiskit* and *qsimcirq* interface in the CuQuantum Appliance (container different from the CPU and GPU container),
[k] Open source version benchmarked, closed source versions available on cloud, [l] Closed CPU and GPU version (GPU not accessible)
[m] Available in CUDA Quantum that uses CuQuantum simulators, [n] Derived from ProjectQ,
[o] Measurements can be derived from probability of a qubit in a particular state, [p] Fundamental design allows for the differentiation of
variational parameters, [q] Support for shots in different providers, [r] Not the typical density matrix support, [s] Q#.

### Performance evaluation procedure

One of the features of our toolchain is the translation of the high-level OpenQASM instruction set of a given quantum algorithm to the specific instruction set of the chosen software package. The translation process thus allows for the introduction of auxiliary function calls which capture the resource consumption of computations. An example for *Cirq* is shown in Fig. 2. Here, the translator writes the appropriate quantum instruction set into a file, simply bracketing it with two additional timer commands. The files generated by the translator, hence capture the effective Time-to-Solution (TtS) or wall-clock time of a computation, which we use to gain insights into the performance characteristics of the particular package in a given configuration.

Further, we note that it is possible to capture other per-formance characteristics like the memory consumption by complementing the time counters with relevant memory counters, e.g. using the package *memory-profiler* in Python or the *Profile* module in Julia, but extraction of the actual resource usage becomes non-trivial in multi-core and -node HPC settings. We generally keep any additional configuration flags in the software packages at their default value and do not tune settings to the simulation task at hand. However, an exception to the above is that we toggle the precision flag where available for a better compartmentalization of the packages.

We now briefly introduce the hardware of our local HPC cluster, while also noting the limits for our benchmarks resulting from constraints to the memory and time available to its users. The Merlin6 cluster at the Paul Scherrer Institute (PSI) provides access to both CPU and GPU architectures.

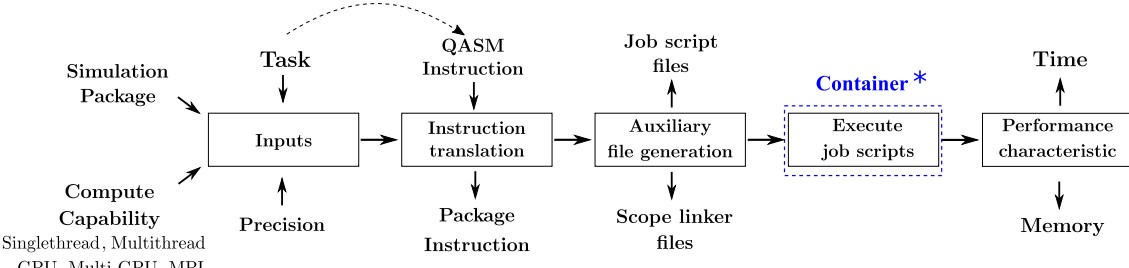

**Figure 1.** Workflow of the toolchain which takes four inputs: (i) Quantum Task, (ii) Simulation package, (iii) Compute capability of the HPC, (iv) Float precision. The task is input as the QASM instruction set which is processed by the translator to further generate the package specific instruction set. Post translation, other auxiliary file generation is triggered that creates the necessary job scripts, linker files that scope the translated run files into the environment of the container. In addition, a script to trigger the job submission onto the cluster is also generated. Finally, the performance characteristics either corresponding to the time or memory are collected for further analysis. (*) The translated files generated from OpenQASM are executed using the package installed on the container, while the pre- and post-processing steps are executed on the HPC node independent of the container.

The CPU architecture consists of Intel processors with each node comprising two sockets that each contain 22 cores. Further, each core supports hyperthreading with two physical threads per core. Regular nodes are equipped with a memory of 384 GB. The support for GPU architectures includes a variety of GPU cards from Nvidia. For the purpose of our benchmark, we focus on the A100 node that comprises of 8 GPUs, each equipped with a memory of 40 GB and total CPU memory of almost 1 TB. To account for the constraints imposed on users, we introduce a fixed set of limits for our benchmarks, which we keep the same for all packages: The maximum run time for a job script is set to 23 hours with a memory limit of 300 GB for the CPU runs and 320 GB GPU/900 GB CPU memory for the GPU runs. This implies that some of the limits in qubit numbers that we report are dictated by the benchmarking environment and should not be understood as an intrinsic limitation of the software package tested. The limitations, for a particular choice of a quantum algorithm are presented in detail in Tab. 3.

## OpenQASM

```
// This is a comment
OPENQASM 2.0;
include "qelib1.inc";

// Insert the time counter, tᵢ   ▷

qreg q[6];
rz(pi*0.1) q[0];
rx(pi*0.2) q[1];
ry(pi*0.3) q[2];
z q[3];
y q[4];
.
.
.
rz(pi*1.5) q[3];
x q[4];

// Inset the time counter, t_f   ▷
// Save the difference, t_f - t_i   ▷
```

## Cirq

```
# This is a comment
from time import perf_counter
import numpy as np
import cirq

t_s = perf_counter()

q = cirq.LineQubit.range(6)
cir = cirq.Circuit()
cir.append(cirq.rz(np.pi*0.1).on(q[0]))
cir.append(cirq.rx(np.pi*0.2).on(q[1]))
cir.append(cirq.ry(np.pi*0.3).on(q[2]))
cir.append(cirq.Z(q[3]))
cir.append(cirq.Y(q[4]))
.
.
.
cir.append(cirq.rz(np.pi*1.5).on(q[3]))
cir.append(cirq.X(q[4]))

simulator = cirq.Simulator()
result = simulator.simulate(cir)

t_e = perf_counter()
np.save('time_perf.npy', t_f-t_e)
```

**Figure 2.** An operational example of the translator, translating the openQASM instructions into the instruction set of the package, *cirq*. The translator incorporates timer functions at the start and the end of the circuit execution.

## Tasks used for benchmarking

As noted earlier, the overall wavefunction describing $N$ qubits can be represented as a statevector with $2^N$ complex entries. Operations defined on $m$-qubits that update this statevector, while preserving its norm, are formally represented by $2^m \times 2^m$ sized matrices, $U_m$, which satisfy the unitary condition $U_m U_m^\dagger = I$, with $I$ being the identity matrix. It should be noted that gates involving only $m$ qubits at a time, can be applied onto a state vector in $O(2^m \times 2^N)$ operations. In quantum computing, such unitaries are generally referred to as quantum gates, which serve as the building blocks of the QASM implementation of each compute task.

With the unfavorable, exponential resource scaling of both statevector and unitaries with increasing qubit numbers, it is not necessarily optimal to perform matrix-vector multiplication directly. Many simulation packages therefore exploit strategies that involve sparse matrices and update the statevector based on indexing, circumventing the need for an explicit storage of the operators during multiplication[79].

We chose three paradigmatic tasks to benchmark our selected set of packages. They are motivated by typical use-cases, such as the simulation of the dynamics of an arbitrary Hamiltonian through digital quantum gates generally prescribed by a Trotter approximation[80], the random circuit

sampling experiments used to establish the transition to a regime of quantum advantage[81], and the Quantum Fourier Transform as a core algorithmic component to many applications[6].

## Task 1: Dynamics of the XYZ-Heisenberg model

In this task, we follow the prescription for a circuit implementation of the dynamics of the Heisenberg model as devised by Smith et al.[82]. We consider a spin-1/2 chain with open boundary conditions, with the interaction given by the Hamiltonian

$$H = -J_x \sum_i \sigma_x^i \sigma_x^{i+1} - J_y \sum_i \sigma_y^i \sigma_y^{i+1} \\ -J_z \sum_i \sigma_z^i \sigma_z^{i+1} + h_z \sum_i \sigma_z^i. \tag{1}$$

where $J_{\{x,y,z\}}$ indicates the interaction strengths with the nearest neighbors, $h_z$ the strength of the magnetic field and $\sigma_x$, $\sigma_y$, $\sigma_z$ are Pauli matrices given by

$$\sigma_x = \begin{bmatrix} 0 & 1 \\ 1 & 0 \end{bmatrix}, \sigma_y = \begin{bmatrix} 0 & -i \\ i & 0 \end{bmatrix}, \sigma_z = \begin{bmatrix} 1 & 0 \\ 0 & -1 \end{bmatrix}.$$

The dynamics of the Hamiltonian is captured by the unitary $e^{-iHt}$, can be approximated by series of unitaries obtained by Trotter-Suzuki expansion[83] as outlined in Ref.[82]. This results in $M$-unitaries given by

$$e^{-iHt} = (e^{-iH\Delta t})^M \tag{2}$$

with $\Delta t = t_f/M$, where $t_f$ is the total time of evolution and $M$ the level of discrete "Trotter steps". Each unitary of the form $e^{-iH\Delta t}$ can be further broken down into to

$$e^{-iH\Delta t} = \prod_{j \text{ even}} A_j \prod_{j \text{ odd}} A_j \prod_j B_j + O((\Delta t)^2) \tag{3}$$

where

$$A_j = e^{-i(-J_x \sigma_x^i \sigma_x^{i+1} - J_y \sigma_y^i \sigma_y^{i+1} - J_z \sigma_z^i \sigma_z^{i+1})\Delta t} \tag{4}$$

and

$$B_j = e^{-i(h_z \sigma_z^i)\Delta t}. \tag{5}$$

In the digitized simulation of the evolution, the Hamiltonian term $A_j$ can be mapped to a unitary $N(\alpha, \beta, \gamma)$ given by

$$N(\alpha, \beta, \gamma) = e^{i(\alpha \sigma_x \otimes \sigma_x + \beta \sigma_y \otimes \sigma_y + \gamma \sigma_z \otimes \sigma_z)} \tag{6}$$

where $\alpha = J_x \Delta t, \beta = J_y \Delta t, \gamma = J_z \Delta t$ encode the respective interaction strength and time steps. This unitary can then be optimally represented using a sequence of single qubit and two qubit Controlled-NOT (CNOT) gate operations as outlined in the Ref.[82] (optimally here refers to a decomposition with minimum number of CNOTs). We simulate the dynamics of the XYZ-Heisenberg Hamiltonian to a final time $t_f = 1$ with a stepsize of $\Delta t = 0.01$ by setting the initial state to $|0\rangle^{\otimes N}$ with $|0\rangle = \begin{bmatrix} 1 & 0 \end{bmatrix}^T$ and $J_x = 1, J_y = J_z = h_z = 0.1$.

## Task 2: Random Quantum Circuits

One of the first experiments that established claims of quantum advantage was the Random Quantum Circuits (RQC) sampling experiment implemented on the Sycamore chip[84]. In our benchmarks, we have used QASM files (specifically, the set marked as the EFGH pattern) supplied by the authors of the original experiment.

The usage of this fixed set of QASM files ensures fair benchmarking across different simulation packages as the generation of RQC circuits involves randomization, which may have an impact on circuit runtime. Central to the RQC experiments are the single qubit gates given by the Pauli $\sigma_x$, $\sigma_y$ and

$$\sigma_w = (\sigma_x + \sigma_y)/\sqrt{2} \tag{7}$$

as well as the hardware-specific two-qubit gate

$$\text{fSim}(\theta, \phi) = \begin{bmatrix} 1 & 0 & 0 & 0 \\ 0 & \cos(\theta) & -i\sin(\theta) & 0 \\ 0 & -i\sin(\theta) & \cos(\theta) & 0 \\ 0 & 0 & 0 & e^{-i\phi} \end{bmatrix}. \tag{8}$$

The general scheme of RQC involves a sequence of "blocks" of gates, with each block consisting of single qubit gate randomly chosen from $\{\sqrt{\sigma_x}, \sqrt{\sigma_y}, \sqrt{\sigma_w}\}$ for each qubit and a two qubit unitary which can be decomposed into gates involving $\sigma_z$-rotations and the fSim$(\theta, \phi)$ gate where the parameters $\theta, \phi$ depend on the pairing of the qubits on which the fSim gate acts[84,85].

Recently, the claim of quantum advantage in the original experiment has been challenged as tensor networks deployed on HPC clusters were able to reproduce the expected linear cross-entropy benchmarking fidelity associated with original experiment[86]. However, the boundaries keep moving as more efficient classical simulation methods as well as the quantum hardware[87] continue to be developed, making this - despite the lack of concrete applications - a highly relevant field for benchmarking[81,88].

We note that our benchmarks involving this task start at a system size of 12 qubits, as this was the minimum number for which QASM circuits were supplied in the original 2019 publication[84] with the circuit depth varying almost linearly in the number of qubits.

## Task 3: Quantum Fourier Transform

The Quantum Fourier Transform (QFT) is a key ingredient in many quantum algorithms, such as Shor's factoring algorithm[89], and - as a paradigmatic circuit element - has already been benchmarked in other works in the literature (e.g. Refs[90,91]). Our inclusion of QFT thus provides a reference point for comparisons to previous and future results. Additionally, properties of the final state obtained after the application of QFT can be directly calculated analytically with the expressions below, and therefore be used to validate the benchmark results in absolute terms.

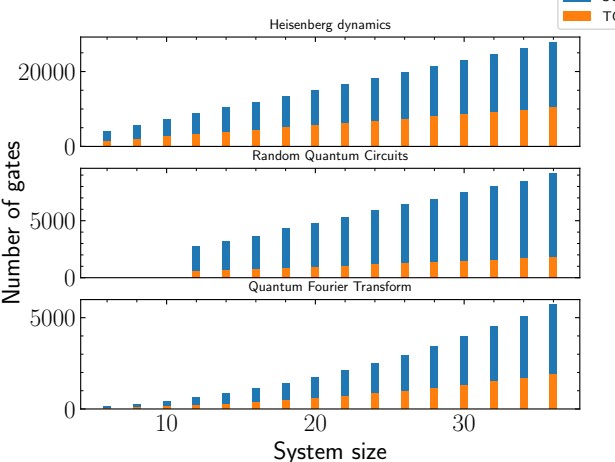

**Figure 3.** Scaling of the number of gates for each task as a function of system size. The number of single qubit gates (SQG) and two qubit gates (TQG) are illustrated as a very simple proxy for the complexity of the calculations that have to be performed.

QFT maps the state $|\psi\rangle = \sum\limits_{i=0}^{M-1} \alpha_i |i\rangle$ where $M = 2^N$ bitstrings (basis states) to $|\phi\rangle = \sum\limits_{i=0}^{M-1} \beta_i |i\rangle$ where $\beta_k = \frac{1}{\sqrt{M}} \sum\limits_{m=0}^{M-1} \alpha_m \omega_M^{mk}$ with $k = 0, 1, 2, 3, ..., M-1$ and $\omega_M = e^{\frac{2\pi i}{M}}$. The corresponding unitary is given by

$$QFT_N = \begin{bmatrix} 1 & 1 & 1 & 1 & \dots & 1 \\ 1 & \omega_M & \omega_M^2 & \omega_M^3 & \dots & \omega_M^{M-1} \\ 1 & \omega_M^2 & \omega_M^4 & \omega_M^6 & \dots & \omega_M^{2(M-1)} \\ 1 & \omega_M^3 & \omega_M^6 & \omega_M^9 & \dots & \omega_M^{3(M-1)} \\ \vdots & \vdots & \vdots & \vdots & \ddots & \vdots \\ 1 & \omega_M^{M-1} & \omega_M^{2(M-1)} & \omega_M^{3(M-1)} & \dots & \omega_M^{(M-1)(M-1)} \end{bmatrix}.$$

QFT can be represented using Hadamard and controlled phase gates acting on an initial state $|\varphi\rangle = |\varphi_0 \varphi_1 \varphi_2 ... \varphi_{n-1}\rangle$, where $\varphi_i = \{|0\rangle, |1\rangle\}$ resulting in the state

$$QFT(|\varphi\rangle) = \frac{1}{\sqrt{M}} \Bigl\{ \Bigl( |0\rangle + e^{2\pi i[0.\varphi_{n-1}]} |1\rangle \Bigr)$$
$$\otimes \Bigl( |0\rangle + e^{2\pi i[0.\varphi_{n-2}\varphi_{n-1}]} |1\rangle \Bigr) \otimes ....$$
$$\otimes \Bigl( |0\rangle + e^{2\pi i[0.\varphi_0 \varphi_1 .... \varphi_{n-2}\varphi_{n-1}]} |1\rangle \Bigr) \Bigr\} \qquad (9)$$

where $[0.\varphi_0 \varphi_1 ... \varphi_l] = \sum\limits_{p=0}^{l} \varphi_p 2^{-p}$. The $\sigma_z$ expectation value of Eq. (9) at any site $i$ will be equal to zero, which we use to validate the computations and order the benchmarked packages based on their precision via their proximity to zero, as detailed in a later section.

### Circuit Structure and Gate Count

In Fig. 3 we illustrate the gate count and circuit composition in terms of one- and two-qubit gates as a function of system size. The Heisenberg dynamics simulation (Task 1) involves a sequence of single qubit and two qubit nearest neighbor gates that are 2-local due to the nearest neighbor description of the Hamiltonian. The scaling of the number of gates is linear in the system size, i.e. the number of qubits $N$. It also depends on the final time of the dynamics simulated, which is however held constant in this task. For the case of RQCs the unitaries are realized on a 2D grid (matching the Sycamore architecture), interspersed with permutations between interaction "blocks". The gates are one- and two-body gates and their number scales linearly with the total number of qubits. For the chosen depth of the circuits the total number of gates is about a factor three or four smaller than the Heisenberg dynamics task and the fraction of two-qubit gates is also lower. Finally, for the case of QFT, the scaling of the unitaries follows a quadratic scaling $\propto N^2$ with a lower number of overall gates than the other tasks.

## Results

Our results focus on the wall-clock time recorded for each combination listed in Tab. 2. It captures the elapsed time for the application of the circuits on the starting state and does not involve the calculation of expectation values.

We present the results first for the three tasks using a single computational core of a single node on our compute cluster, while treating the single and double precision cases separately. We then study the time to solution for a set of packages and a given task as a function of the problem size, i.e. the number of qubits $N$. Given that that the unitaries in the considered circuits are one- and two-site gates only, we expect an optimal implementation to scale as $O(N_{\text{gates}} \times 2^N)$, where $N_{\text{gates}}$ denotes the number of gates of the circuit (which itself depends linearly or quadratically on $N$).

Interestingly we observe for many packages that this asymptotically expected behaviour only sets in for relatively large qubit numbers, while for smaller number of qubits the time to solution has a substantially weaker system size dependence, and can even be almost system size independent in some particular cases. We suspect that a possible origin of this behaviour could be internal circuit analysis and optimisation by certain packages, whose overhead only starts to amortize for the larger system sizes.

While the small system scaling can vary very significantly between packages, we focus on the exponential scaling part to extract a ranking between packages, representative of their efficient handling of circuits for large numbers of qubits.

As illustrated in Fig. 4, we fit the time to solution as a function of qubits for a given package and task to the following formula in the large $N$ regime:

$$t = \exp(a) \times \exp(b)^N,$$

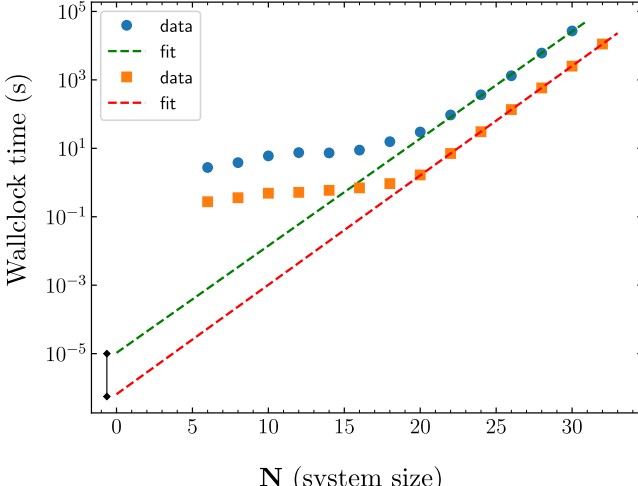

**Figure 4.** Extraction of relative speed and scaling overhead. Performance time scales linearly in the large-$N$ limit on the log scale i.e., $\log(t) = a + bN$ where $t$ is the wall-clock time and $N$ is the system size (number of qubits). We extract the $y$-intercept, $a$ and slope, $b$ using the above linear fit, which we use for relative speed assessment (marked by the black solid line around $N = 0$).

or in logarithmic form:

$$\log(t) = a + b \times N.$$

The two fit parameters $a$ and $b$ are interesting for the purpose of comparing the implementations and the performance. The parameter $b$ in an efficient implementation is expected to be simply $b = \log(2)$[*], while the parameter $a$ captures the prefactor via $\exp(a)$, and turns out to be the main proxy for the performance difference among the various packages.

In the current analysis, we restrict the extraction of $a$ and $b$ among different packages to the case of singlethread performance but it can be similarly extended to other compute capabilities, provided enough data points are available in the regime of exponential scaling. For comparison across other compute capabilities we simply use wall-clock time ratios.

The results figures in our manuscript show our aggregated findings, which can make them hard to read with regards to individual packages. To allow for a better data exploration, we have therefore set up an interactive website at https://qucos.qchub.ch, where readers are invited to perform targeted analyses across all the benchmarked dimensions and packages.

### Singlethread performance

In this part of our study, we configure the toolchain to limit the available compute capability to a single computational thread. The aim is to investigate the effect of numerical precision

(single vs. double) on both the wall-clock time scaling as well as the maximum system size that can be simulated. While this difference of 32 bit vs. 64 bit representation will not allow for a much larger number of qubits (a factor two reduction in total memory), less data will have to be processed at every step, allowing for speed-ups depending on the package internals.

Figs. 5 and 6 show the recorded wall-clock time as a function of the number of simulated qubits for single or double precision singlethread configurations, respectively. We also plot the ratio of each package's wall-clock time with that of the fastest (smallest $a$ coefficient) large-$N$ simulator, more easily revealing distinct regimes of scaling among the packages. This is most pronounced in the single precision case, where the wall-clock time can differ by up to a factor of 1000 for system sizes around $N = 26$ qubits.

We start by noting that in the top row of Figs. 5(a) and 6(a), we observe that some packages exhibit almost straight lines for the entire range of system sizes, while other packages have a rather small slope at lower numbers of qubits and only cross over to the steeper slope for qubit numbers of about 20. This leads to the paradoxical situation that the fastest package in Fig. 6(a) at the largest qubit numbers (*qiskit*) is among the slowest packages for eight qubits [†].

Among the packages that support single precision, shown in Fig. 5(a), we can clearly identify that the package *qsimcirq* performs best in the large-$N$ limit. For double precision resolution, shown in Fig. 6(a), the best performing package is less clear, but *qiskit* and *qpanda* are consistently among the fastest packages (*qsimcirq* only supports single precision). However, in the small-$N$ limit, below $N \sim 15$ qubits, many other packages perform better in comparison to these two, indicating smaller constant computational overhead. In particular, the packages *yao* or *qrack* show very little overhead at low $N$, scaling exponentially from the start, which can provide a benefit for small system size simulations. In the other extreme, the package *hybridq* has a high computational overhead for small and intermediate qubit numbers, yet almost competes with *qsimcirq* for large $N$.

While the behavior of constant computational overhead followed by a exponential scaling in the time is almost consistently observed across the different tasks, we note that the wall-clock time in the case of QFT is small in comparison to the Heisenberg dynamics and Random Quantum Circuits, which we partially attribute to the lower number of gates required at a given system size (cf. Fig.3).

In Figs. 5(b) and 6(b) we quantify speed and computational overhead at large scales for the various packages using fit parameters $a$ and $b$. As we are unable to obtain reliable fit parameters for the package *hybridq* due to the paucity of data in the large-$N$ limit, we exclude it from the comparison.

The ratio of the exponentials $e^a$ indicates the approximate wall-clock time overhead incurred for each package in comparison to the best performing one. In the single precision

---

[*]with some small enhancement to compensate for the fact the the number of gates depends itself on $N$.

[†]A similar observation holds for *hybridq* in the single precision case, although it is never the fastest package, even though it gets quite close.

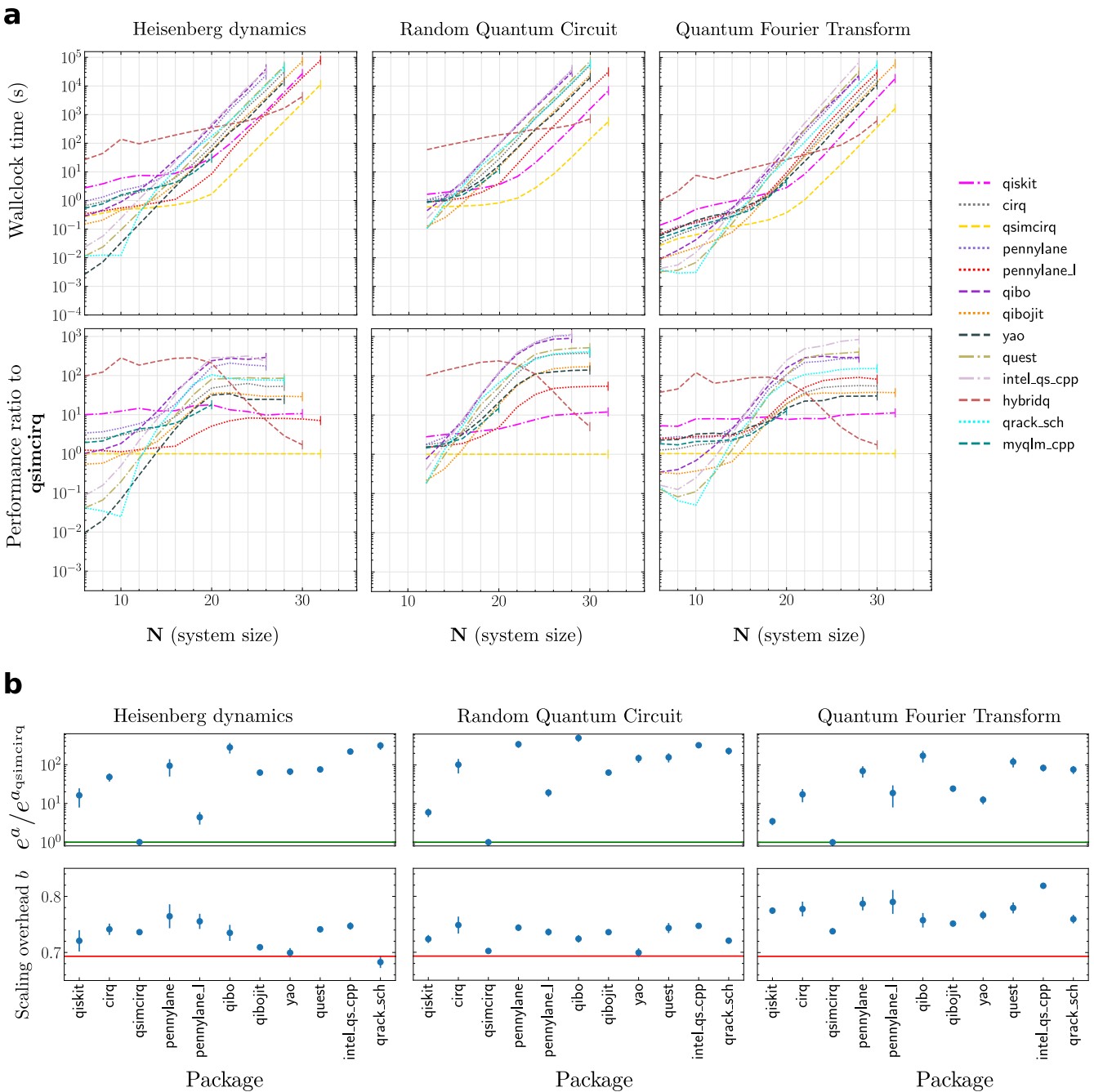

**Figure 5.** (**a**) Singlethread performance (single precision). (*top*) Absolute wall clock time as a function of qubit number for packages that support single precision according to the package documentation. (*bottom*) Performance ratio with respect to best performing package, here *qsimcirq*. (**b**) Scaling behavior of singlethread single precision performance by extracting *a*, *b* as detailed in the main text. (*top*) Relative slow-down with respect to the fastest large-*N* package, *qsimcirq*. *(bottom)* Deviation from the expected log(2) exponential scaling (red line) as a function of the number of qubits.

setting we notice that ranking remains almost same with minor deviations across the different tasks. In contrast, the double precision setting does not follow a strict trend as we notice varying performance with respect to different tasks. In terms of the computational overhead captured by *b*, which is ex-

pected to be close to $\log 2 \sim 0.7$, we find that some packages get close to this theoretical limit in case of the Heisenberg dynamics and RQC, yet the overhead in QFT circuits consistently appears to be larger. This might be due to the number of gates of the circuits, which depend quadratically on *N*, and

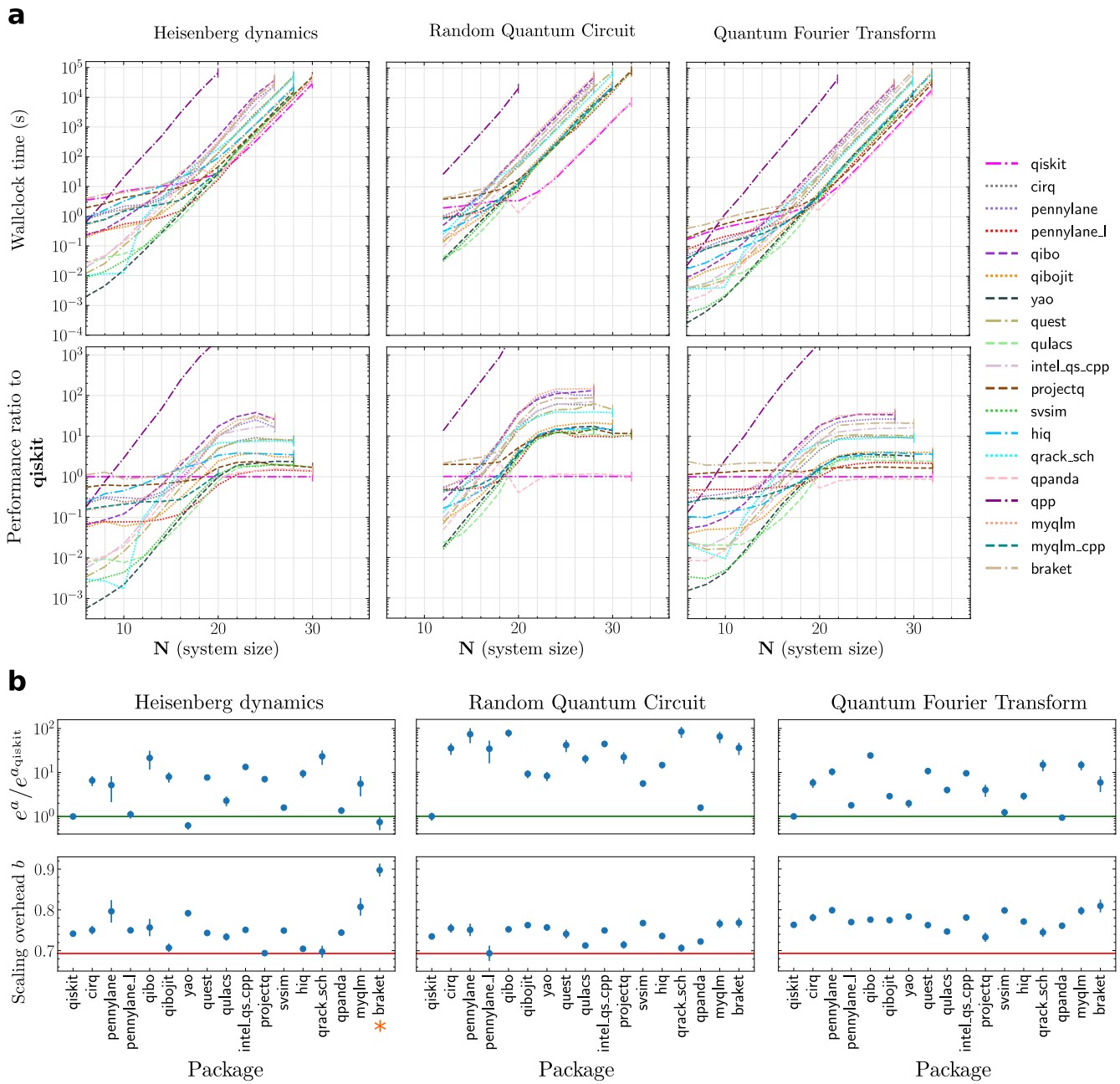

**Figure 6.** (**a**) Singlethread performance (double precision). (*Top*) Absolute wall clock time as a function of qubit number for packages that support double precision according to the package documentation. (*Bottom*) Performance ratio with respect to optimally performing package, here *qiskit*. Note that numbers above one correspond to a relative slow-down. (**b**) Scaling behavior of singlethread double precision performance by extracting *a*, *b* as detailed in Fig. 4 for different quantum algorithms. (*Top*) Relative speed-down with respect to the reference package, *qiskit*. (*Bottom*) The red line indicates the expected scaling following $\log(2)$ as a function of the number of qubits. The paucity of data points in the large *N*-limit for the package myqlm_cpp limit the extraction of *a* and *b*. * In the an effort to minimze the errors on *a* and *b* for package braket, the final data point has been excluded while fitting.

lead to an apparent enhancement of our parameter *b*.

## Impact of parallelization strategies

In this section, we compare the performance of various simulation packages across the three tasks under different hardware architectures available on our HPC cluster. Specifically, we

focus on the observed performance differences between singlethread, multithread and GPU compute capabilities. We briefly outline the multithreading and GPU compute options as declared in the job scripts. For multithreading, the number of threads utilized is 84 (42 cores with hyperthreading turned on, i.e. two physical threads per core). For the case of GPU, we use the Nvidia A100 compute node while varying the number of GPUs from 1 to 8. The results with a single GPU are presented here while the ones with multi-GPU are presented in the online data explorer at `https://qucos.qchub.ch`.

Overall, the results shown in Fig. 7 again show the familiar scaling behavior of an almost constant computation overhead at small system sizes, which changes to an exponential scaling at a particular numbers of qubits. The transition point between the different regimes of scaling notably shifts to larger system sizes by 2-4 qubits, depending on the package, when moving to multithreading and GPU-based calculations. As a consequence, correspondingly larger system sizes can be simulated in the same fixed time. In the Heisenberg model and QFT this shift to larger accessible system sizes is most apparent in Fig. 7. However, at smaller system sizes before the transition point, a GPU-associated overhead penalty often makes calculations slower that even its singlethread counterparts. The packages *Yao* and *QUEST* are two notable exceptions here, at least in the RQC and QFT tasks. Notably, in the RQC tasks, the two leading packages at large system sizes, *Qiskit* and *QPanda*, are notably faster than other simulators. The package *cuquantum* only shows the onset of exponential scaling in the Heisenberg model simulation task. It otherwise retains almost constant performance for all system sizes in the RQC and QFT tasks, with the size limit of $N = 32$ coming from our memory and not the time limit.

**Speedup trends**

As evident from the analysis so far, the benefit of using a more powerful compute capability can vary based on task and system size. To better judge where significant speedups can be obtained within one and the same package, we therefore plot the respective performance ratios for each package individually in Fig. 8. As not all packages support all hardware capabilities (esp. GPU) some entries are left blank, but the majority supports both single- and multithread computations, exhibiting performance gains of factors between 5 and 30 for some intermediate problem sizes $N > 20$. GPU performance enhancements are even more striking, reaching up to factors of 500 for the package *Quest*. *pennylane*'s C++ version also exhibits significant performance gains in the large $N$ limit. Interestingly, the package *yao* instead gains the most in small and intermediate problem sizes, hinting at a fundamental difference between its CPU and GPU implementation.

**Cross-validation of results**

In this section, we present results that we use to both validate our toolchain implementation as well as asses the precision of the benchmarked statevector simulators.

Calculating a reference solution for the Heisenberg dynamics with high accuracy remains a challenge as there is no exactly solvable analytical form. RQC, by definition, has no analytical form. However, in the case of QFT as outlined earlier, the final statevector obtained after the application of the QFT has a closed form i.e., it can be estimated analytically, via Eq. 9. As previously noted, the $\sigma_z$ expectation value of the state vector after the application of the QFT has to remain zero at all sites. This observation allows us to validate the precision settings of the different packages and also to order them based on the accumulated error. In Fig. 9, we plot the quantity $\log_{10}(\sum_i |\langle \sigma_z^i \rangle|)/N$ for different simulation packages and note that all packages are in good agreement with the expected result of 0, thereby validating the entire toolchain for the given task. To validate the precision settings, we expect the above defined quantity for a package that promises single precision to be on the order of 7 decimal digits, while those that promise double precision to be on the order of 16 decimal digits. We infer that the packages *qiskit* and *pennylane* despite promising single precision via an optional setting actually continue to operate in double precision.

For the tasks of Heisenberg dynamics and RQCs the validation of the toolchain and assessment of the quality of the solution remains a challenge. We therefore employ a cross-validation of results by computing the quantity

$$\Delta \text{Expectation}(p_1, p_2) = \log_{10}(\sum_i (|\langle \sigma_z^i \rangle_{p_1} - \langle \sigma_z^i \rangle_{p_2}|)). \quad (10)$$

This measure is inspired by the case of QFT, but instead of comparing the $\langle \sigma_z \rangle$ at all sites to the expected result, we compare among the packages and validate the quality of the solution if the value of the quantity as defined in Eq. 10 is on the order of the precision target set in the configuration. We note that the above quantity involves computation of the expectation value of $\sigma_z$ at all sites, which we choose to compute for a system size of $N = 16$ as the mechanism deployed to compute the expectation values remains the same irrespective of the system size. Significant deviations in the calculated expectation values from the results of the other packages points to a potential defect in some component of the toolchain specific to the chosen simulator package, including the package itself. In addition, any deviation in the performance characteristics from the general trend also implies a potential failure either in the toolchain or the packages itself demanding further investigation. We note that we have observed such a deviation in the general trend only for a single package, namely *QCGPU* which apparently performs much faster for a system size of $N = 32$, yet fails to produce results without giving any error message in our setup.

In Fig. 10, we present matrix plots with packages on rows and columns representing the $p_1$ and $p_2$ as in the Eq. 10. We clearly notice that for the case of single precision the packages that differed from the single precision as in the case of QFT, see Fig. 9, reflect similar behavior across the different tasks i.e., *qiskit* on different architectures and *pennylane(py)*

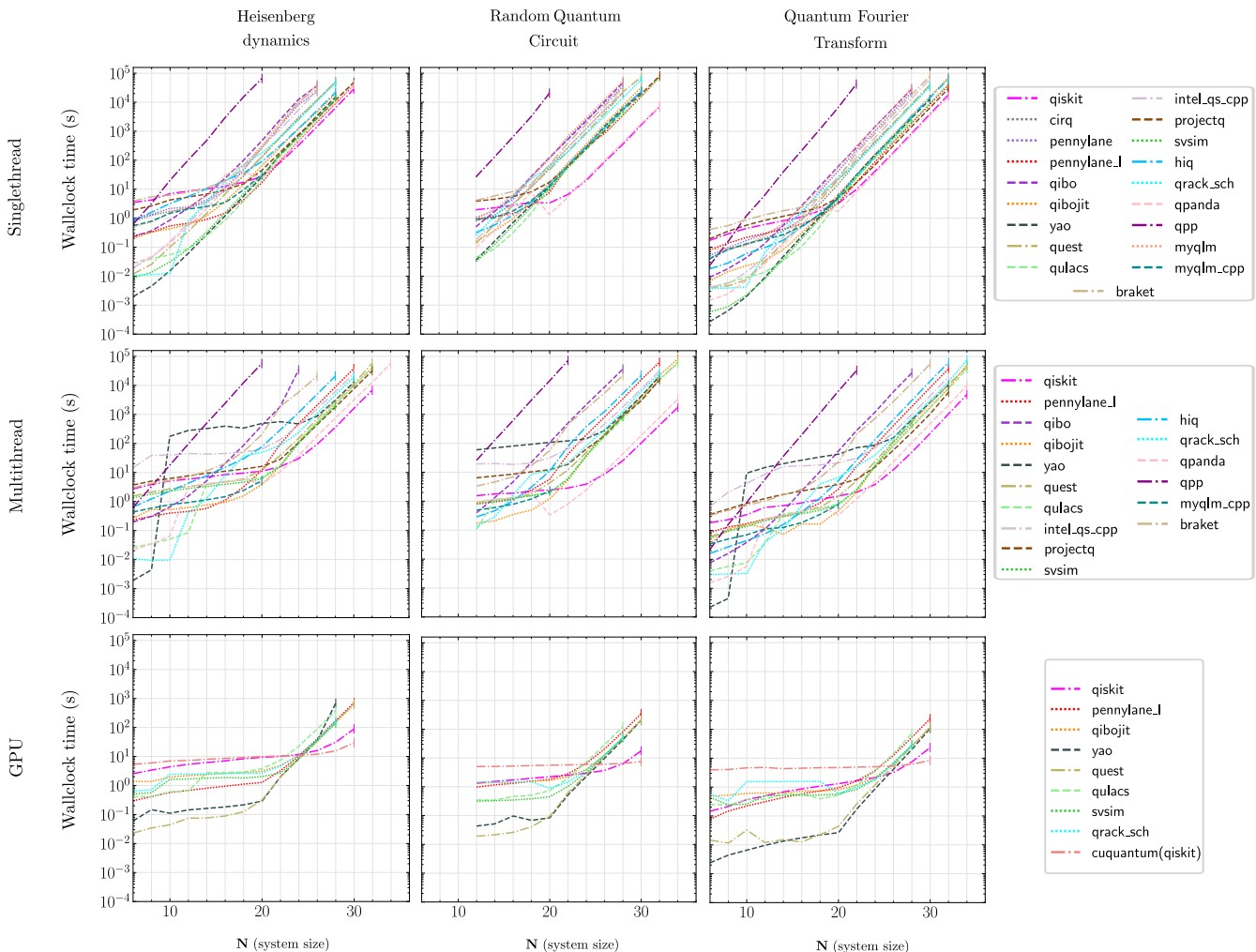

**Figure 7.** Performance comparison of simulation packages supporting double precision across different hardware architectures.

operate on double precision even when the single precision flag is set. In addition, RQC circuits when run using *pennylane* set to double precision on the GPU architecture fail to produce a solution that is in agreement with the other packages. To further validate the above observation, we compare both, the generated run files and the job scripts related to the package and note that the generated files vary only in the choice of the hardware and are equivalent elsewhere. Due to the above, we conclude that *pennylane* with double precision on the GPU architecture fails to produce the correct.

**Limitations of the benchmarked quantum simulators**
Three main types of limitations were found, which we classify as design limited, time limited and memory limited. Design limitation refers to the inability of the simulation package to go beyond a certain system size due to design decisions considered as a part of the development of the package. Design decisions range over a wide spectrum that include: the choice of the language used to develop the package, the choice of

the core support libraries (for instance: numpy, scipy, AVX acceleration among others), design features to support multi-compute capabilities and so on. Time limited and memory limited are more subjective to the cluster on which the performance is being benchmarked. In our case, we limited the time to a day and the memory to 300 GB on the CPU node and 320 GB-GPU/900 GB-CPU on the GPU node.

Entries in Tab. 3 mark limits and other issues we encountered in simulating the dynamics of the Heisenberg model. Here, $D(N)$ denotes a design limitation starting at system $N$, i.e., a statevector starting $N$ cannot be processed by the simulation package even if there is sufficient memory and time resources available. $T(N)$ and $M(N)$ denote the time limitation and the memory limitation starting at $N$. These limits are dynamic and vary subjectively, depending on cluster resource constraints and the benchmarking task at hand. We expect $D(N)$'s to remain the same across various tasks, however, $T(N)$ and $M(N)$ will likely vary slightly as implied

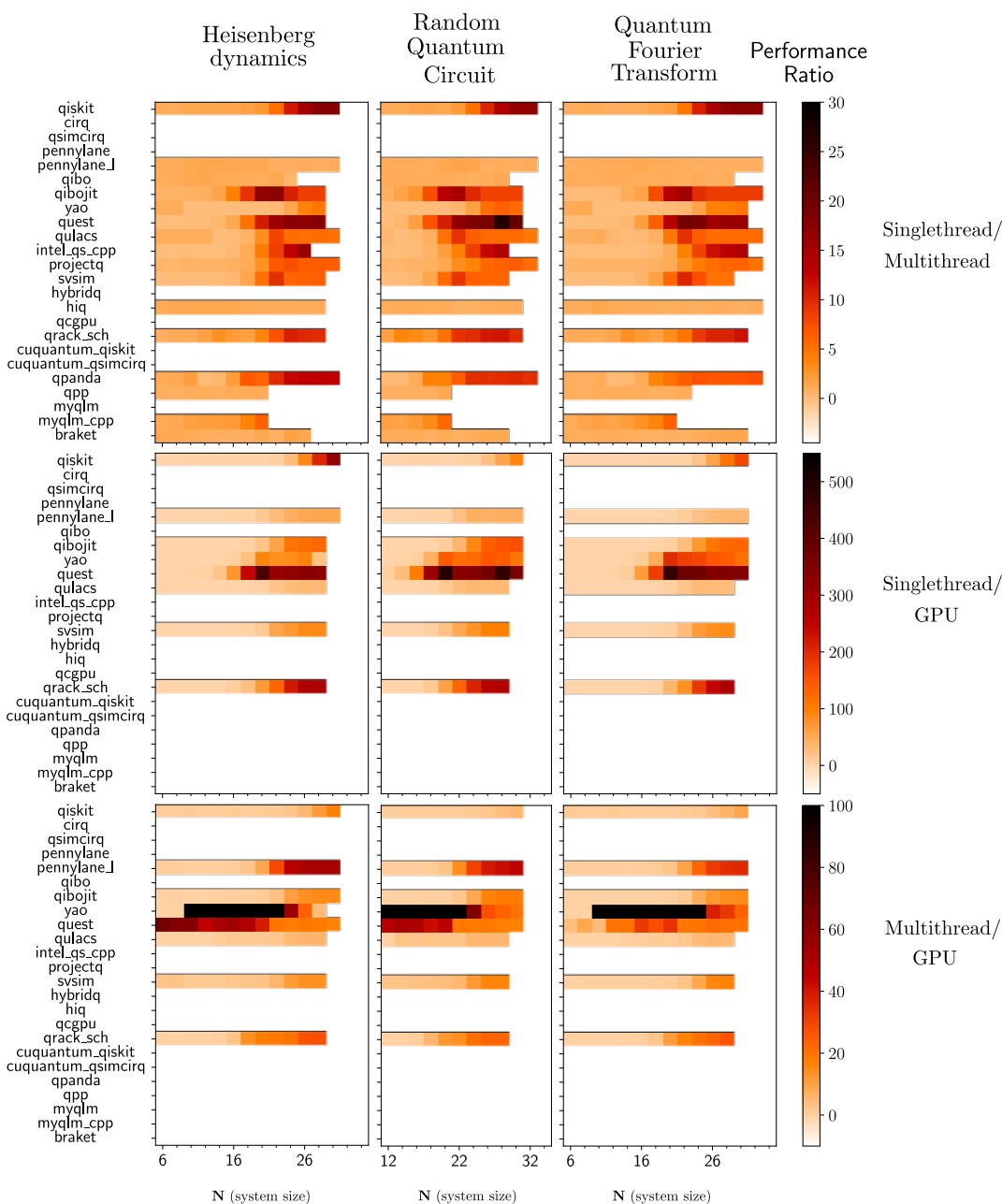

**Figure 8.** Double precision speedup/slowdown due to the increase in computational capabilities with respect to packages supporting double precision as in their documentation as a function of number of qubits. In the large-*N* limit there is a significant speedup with increase in computational resources, captured by the ratios labeled on the extreme right.

by the performance charts shown earlier.

## Outlook

To summarize, we have benchmarked a variety of quantum computing simulation packages that are able to leverage HPC capabilities. We have automated the benchmarking procedure by developing a containerized toolchain solution, that accepts the simulation package, QASM instructions of the quantum algorithm to be benchmarked and the HPC compute capabilitiy

as input, and outputs the performance characteristics. Central to the automation is the translation of the QASM instruction to the instruction set of the chosen simulation package. In the current benchmarks we have chosen algorithms that are static i.e., non-variational in nature. However, the current translators in our toolchain can easily be equipped with classical optimizers and integrated in a loop to be suitable for automating the benchmarking of variational algorithms. One further interesting direction to explore would be to benchmark the

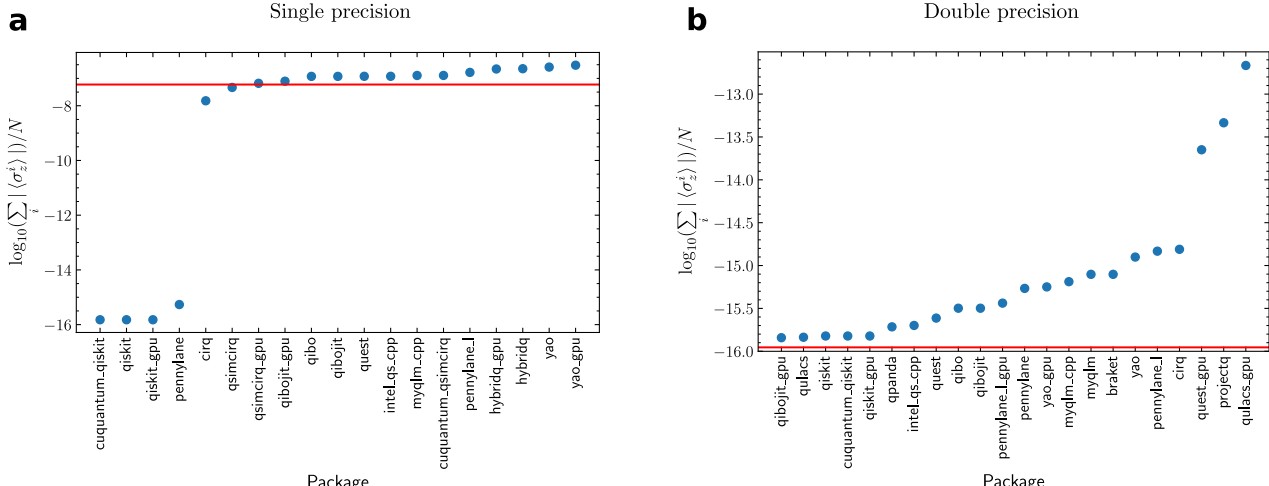

**Figure 9.** Accumulated errors in QFT for N=16 with regards to baseline expectation in (a) single precision and (b) double precision settings. Summed up deviations from the expected $\langle \sigma_z^i \rangle = 0$ are shown for each setting (red line) and packages sorted correspondingly. We note that in (a) packages like *qiskit* and *pennylane* document single precision settings but are found to be running double pression, instead.

**Table 3.** Task dependent feature limitation - Dynamics of the Heisenberg model

| Package | Singlethread | | Multithread | | GPU | |
|---|---|---|---|---|---|---|
| | SP | DP | SP | DP | SP | DP |
| Braket | | $T(28), D(34)$ | | $T(28), D(34)$ | | |
| Cirq | $T(30), D(34)$ | $T(30), D(34)$ | | | | |
| cuQuantum | | | | | | |
| HiQ | | $T(30), M(36)$ | | $T(30), M(36)$ | | |
| HybridQ | $D(32)$ | | $D(32)$ | | $M(30), D(34)$ | |
| Intel-QS(cpp) | $T(28), M(36)$ | $T(28), M(36)$ | $T(32), M(36)$ | $T(32), M(36)$ | | |
| myQLM (py) | | $T(28), D(34)$ | | | | |
| myQLM (cpp) | $RA(22)$ | $RA(22)$ | $RA(22)$ | $RA(22)$ | | |
| Pennylane (py) | $T(28), D(34)$ | $T(28), D(34)$ | | | | |
| Pennylane (cpp) | $D(34)$ | $T(32), D(34)$ | $D(34)$ | $T(32), D(34)$ | $RE$ | $M(32)$ |
| Projectq | | $T(32), M(34)$ | | $M(34)$ | | |
| Qcgpu | | | | | $F(32), M(34)$ | |
| Qibo | $T(28), D(32)$ | $T(28), D(32)$ | $T(28), D(32)$ | $T(28), D(32)$ | | |
| Qibojit | $T(32), M(36)$ | $T(30), M(36)$ | $T(34), M(36)$ | $T(34), M(36)$ | $M(34)$ | $M(32)$ |
| Qiskit | $T(32)$ | $T(32)$ | $M(34)$ | $T(34)$ | $T(34)$ | $T(32)$ |
| QPanda | | $T(32), M(34)$ | | $T(32), M(36)$ | | $NO$ |
| Qrack | $T(30), M(36)$ | $T(30), M(36)$ | $T(34), M(36)$ | $T(32), M(36)$ | $M(32)$ | $M(32)$ |
| Qsimcirq | $D(34)$ | | $D(34)$ | | $M(34)$ | |
| Quest | $T(30), M(36)$ | $T(30)$ | $T(34), M(36)$ | $T(34)$ | | $M(32)$ |
| Qulacs | | $T(32), M(36)$ | | $T(34), M(36)$ | | $M(30)$ |
| Q++ | | $T(22)$ | | $T(22)$ | | |
| SV-Sim | | $D(30)$ | | $D(30)$ | | $D(30)$ |
| Yao | $T(30), M(36)$ | $T(30), M(36)$ | $T(34), M(36)$ | $T(32), M(36)$ | $T(32), M(34)$ | $M(32)$ |

$D(N), T(N), M(N)$ denote design limitation, time limitation and memory limitation respectively.
$F(N)$ represents runs that return success but terminate on short scales defying the expected scaling of time,
*RE* represents *Runtime Error*, *NO* represents *No Output*,
*RA* represents *Restricted Access* (higher system sizes are available via cloud access for a fee).

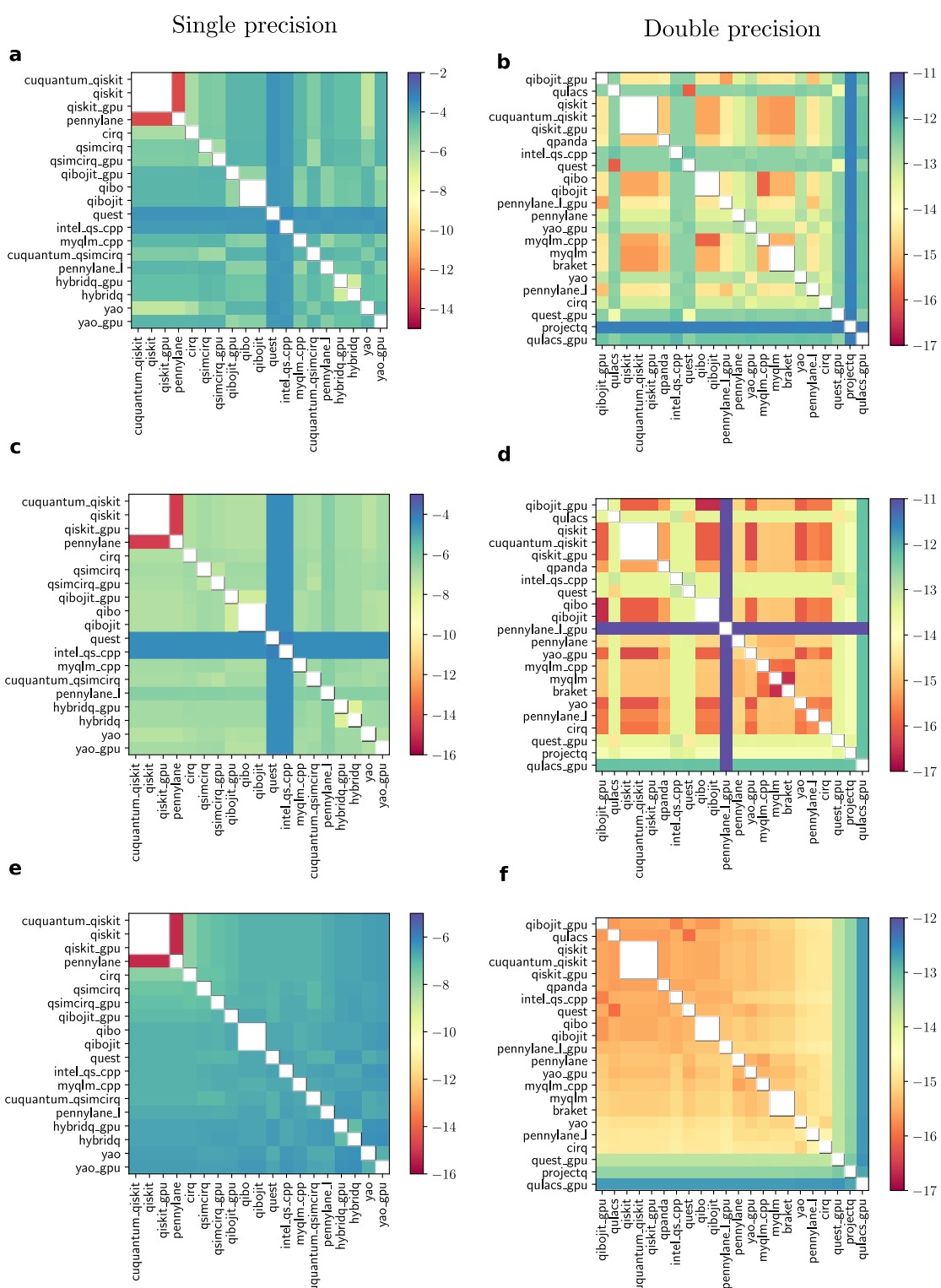

**Figure 10.** Matrix representation of Eq. 10 for simulation packages labeled on the rows and the columns. Each row indicates different tasks benchmarked: (a, b) Heisenberg dynamics, (c, d) RQC, (e, f) QFT. The left column (a, c, e) includes the packages supporting single float precision while right column (b, d, f) includes packages supporting double precision. Some of the exceptions include: *Pennylane* (GPU backend with double precision) is inconsistent, results do not match with the results from other packages. Packages such as *qiskit* (all hardware backends) and *pennylane* which support single precision as in the documentation but are double precision in operation. The white blocks indicate that the data matches exactly.

performance of simulators which support noise and also simulators based on tensor networks. For static algorithms with no parameter updates (non-variational) the toolchain developed in the current context can already be extended to benchmark these kinds simulators.

## Data and Code availability

The entire source code for the QASM translator, toolchain, plotting/data analysis and auxillary files as well as the extracted data can be found at `https://huggingface.co/spaces/amitjamadagni/qs-benchmarks/tree/main`. Additionally, the workflow used for generating the benchmarking results, including the singularity images, the quantum algorithms in the QASM format and sample job scripts can be found at the data repository[92]: `https://zenodo.org/records/10376217`. An interactive web platform for online data exploration and on-demand comparison across packages, tasks and hardware modalities can be found at `https://qucos.qchub.ch`.

## Acknowledgments

We thank the support of the High Performance Computing and Emerging Technologies group at PSI that maintain the Merlin6 cluster, mainly Marc Caubet and Elsa Germann, for providing support on installation issues and crucial insights into the working of the Merlin6 cluster. We further acknowledge funding support by PSI Research Grant 2021-01503.

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
