# Peer review of "Benchmarking Quantum Computer Simulation Software Packages: State Vector Simulators"

_SciPost Physics Core_

## Round 2 · Referee Report · Anonymous (Referee 1) · 2024-8-15

Report

This paper gives an overview of the existing packages that simulate quantum systems. Then it compares the packages with a state-vector simulator with each other. This happens by testing different tasks (quantum circuits) having different features when scaling up the number of qubits. The runtime of the simulations is compared as the number of qubits increases. Also, the accuracy of the simulation is checked by comparing it with known analytical results if they exist or with the rest of the packages. I see that this paper is a good guide for people working in quantum simulations to have an idea about the most suitable simulator for their studies.

"Introduction": This section provides a comprehensive overview of quantum simulations, their applications, and the challenges associated with scaling them up. It delves into the limitations imposed by factors such as runtime, data storage, and processing power as the size of the quantum system increases.

"Quantum computer simulation packages": This section explores alternative quantum simulation methods to state-vector simulation, which is the focus of this article. It discusses tensor network-based approaches, such as Matrix Product States (MPS) and Projected Entangled Pair States (PEPS), detailing their applications and limitations when executed on classical computers. Additionally, it provides an overview of currently available software packages for these simulation methods.

"Containerized toolchain workflow": This section outlines the methodology employed in this study. It provides a high-level overview of the testing procedures without delving into intricate details. Additionally, it introduces the device utilized for the experiments. Comments: 1- Why not use the latest version of some packages such as qiskit? 2- How is it possible that a simulator such as Qrack does not support the typical density matrix and supports noisy simulations? 3- In Figure 2 in the last line of the Cirq instructions the line should probably be np.save('time_perf.npy',t_e-t_s) or change the variable previously defined.

"Tasks used for benchmarking": The tested tasks are clearly defined, along with their associated properties and performance characteristics when the number of qubits increases. Comments: 4- It is not very clear how $\theta$ and $\phi$ depend on the pairing of the qubits for $fSim(\theta,\phi)$. (Last sentence of the third paragraph of "Random Quantum Circuits" subsection)

"Results": The results are presented clearly and well-explained. Comments: 5- Singlethread performance: You mention that the packages yao or qrack show very little overhead at low $N$. But looking at Figure 5a it is not the case for RQC and QFT cases for yao and in Figure 6a in the case of RQC for qrack. 6- Singlethread performance: In paragraph 5 line 2 there is a typo "a exponential" $\longrightarrow$ "an exponential" 7- The colors of the curves of Figure 6a are hard to distinguish. 8- "https://qucos.qchub.ch" is inaccessible. 9- Cross-validation of results: As noted in paragraph 2, lines 18-21, packages like qiskit and pennyLane employ double-precision arithmetic despite operating in single-precision mode. This discrepancy might influence the results presented in Figure 5a, where the performance of these packages is compared against other tools using single-precision calculations.

"Outlook": Comments: 10- End of the last line "extended to benchmark these kinds simulators." $\longrightarrow$ "extended to benchmark these kinds of simulators."

Recommendation

Publish (easily meets expectations and criteria for this Journal; among top 50%)

---

## Editorial Decision

resubmitted